# Partial Vaginectomy, Complete Vaginectomy, Partial Vestibule-Vaginectomy, Vulvo-Vestibule-Vaginectomy and Vulvo-Vestibulectomy: Different Surgical Procedure in Order to Better Approach Vaginal Diseases

**DOI:** 10.3390/ani12020196

**Published:** 2022-01-14

**Authors:** Daniele Zambelli, Simona Valentini, Giulia Ballotta, Marco Cunto

**Affiliations:** Department of Veterinary Medical Sciences, University of Bologna, Via Tolara di Sopra 50, Ozzano dell’Emilia, 40064 Bologna, Italy; daniele.zambelli@unibo.it (D.Z.); simona.valentini@unibo.it (S.V.); marco.cunto@unibo.it (M.C.)

**Keywords:** partial vaginectomy, complete vaginectomy, partial vestibule-vaginectomy, vulvo-vestibule-vaginectomy, vulvo-vestibulectomy, dog

## Abstract

**Simple Summary:**

Total or partial vulvo-vaginectomy or vaginectomy are not routinely surgery due to the complexity of the techniques and because they are considered radical treatments. Furthermore, in literature, there is a paucity of information regarding these techniques and the extent of the reproductive tract resection is not always clearly defined, as the same technique is often named in a different way by different authors, confusing the reader. The aim of this article is to review and describe five surgical procedures on the basis of the correct identification of the anatomical areas: partial vaginectomy, complete vaginectomy, partial vestibule-vaginectomy, vulvo-vestibule-vaginectomy and vulvo-vestibulectomy. For each technique, indications and possible intraoperative and perioperative complications are mentioned. Moreover, authors’ clinical experience in 33 dogs presenting genitourinary lesions not amenable to local resection via simple episiotomy and outcomes are described.

**Abstract:**

Total or partial vulvo-vaginectomy or vaginectomy are not routinely performed due to the complexity of the techniques and because they are considered radical treatments. Little information can be found in the literature, as the same technique is often named in a different way by different authors, confusing the reader. Therefore, the aim of this essay is to describe five different surgical techniques: partial vaginectomy, complete vaginectomy, partial vestibule-vaginectomy, vulvo-vestibule-vaginectomy and vulvo-vestibulectomy. All techniques are described on the basis of the correct identification of the anatomical nomenclature related to structures involved in surgery, in order to give a more precise and unambiguous description and execution of surgical techniques. Moreover, possible intraoperative and perioperative complications and the authors’ clinical experience in 33 dogs are described. All techniques are well tolerated and could be curative in case of benign or malignant tumours that have not yet metastasized and palliative in other cases. Moreover, they are also useful for therapeutic purposes for chronic vaginitis, severe vaginal cysts or congenital abnormalities. It is our opinion that having five different available techniques to approach vaginal disease is useful to perform the best surgery according to the clinical findings, patient’s characteristics, technique invasiveness and whether it is palliative or not.

## 1. Introduction

Many diseases, such as benign or malignant neoplasia, vaginitis, vaginal prolapse and congenital abnormalities, could affect the copulatory organs (vagina, vestibule and vulva) in the dog and most of them could be age-related [1]. For these pathologies, surgical management could require simple or extremely complex approaches [1,2,3,4]. Vaginal surgeries are mostly performed via a caudal approach, and episiotomy is frequently required to improve exposure of vaginal or vestibular abnormalities [1,2,4]: if this approach does not allow a complete resolution of the disease, because of the type, size or location of the lesions, a total or partial vulvo-vaginectomy or a vaginectomy are then requested [1,2,4,5,6,7]. These kinds of surgery are not routinely performed because of the complexity of the techniques and are considered radical treatment [4].

Unfortunately, in literature there is little information regarding the surgical techniques performed on the copulatory organs. In particular, the extent of the reproductive tract resection is not always clearly defined, as the same technique is often named in a different way by different authors, confusing the reader.

In the authors’ opinion, a correct anatomical nomenclature related to structures involved in surgery could lead to a more precise and unambiguous description and execution of surgical techniques.

Therefore, our purpose is to give a more specific and specialized point of view, starting from the Nomina Anatomica Veterinaria (N.A.V.) to describe in detail five different surgical techniques available for the female reproductive system of the dog.

N.A.V. subdivides the female reproductive system in Organa genitalia feminina (ovary, uterine tube, uterus, vagina and vestibule) and Partes genitales femininae externae (vulva, clitoris, feminine urethra) [8]. The vagina is defined as a canal from the cervix to the hymen or vestibule: the hymen is a fold of mucosa just cranial to the transverse plane of the urethral orifice and it is poorly developed in domestic mammals. The vagina opens into the vestibule through the vaginal ostium (ostium vaginae), just cranial to the urethral orifice. The vestibule (vestibulum vaginae) extends from the hymen to the labia: it is deeper in the dog than in humans, it is not considered part of the external organ and includes the urethral tubercle, an elevation bearing the urethral orifice. Vulva and clitoris are the external organs [8,9].

On the basis of the correct identification of the anatomical areas, five surgical techniques can be described: partial vaginectomy, complete vaginectomy, partial vestibule-vaginectomy, vulvo-vestibule-vaginectomy and vulvo-vestibulectomy (Figure 1).

Partial vaginectomy involves the removal of a variable tract of the vagina, always including the cervix. All vaginal tissue, from the cervix to the hymen, are removed in complete vaginectomy. A partial vestibule-vaginectomy is similar to the complete vaginectomy, but a variable part of the vestibule is involved: in this case, a urethrostomy performed in the segment of the vestibule tissue left in situ is mandatory. In a vulvo-vestibule-vaginectomy, all the copulatory organs (vagina, vestibule and vulva) are removed and a perineal urethrostomy has to be performed. In a vulvo-vestibulectomy, the externa genitalia and a variable vestibular length is removed: the urethral tubercle is not involved (Figure 1). Except for the vulvo-vestibulectomy, all surgeries previously described require ovariohysterectomy in intact bitches or hysterectomy in ovariectomized ones.

The objective of this study is to describe, in function of the N.A.V. organs’ description, these five different surgical approaches to the bitch reproductive system as performed by the authors. Since the literature provides scarce and ambiguous explanation about these kinds of techniques, another purpose is to offer to practitioners an exhaustive description in order to provide a repeatable, safe and tolerable surgical procedure. For each technique, indication and possible intraoperative and perioperative complications are mentioned. Moreover, authors’ clinical experiences in 33 female dogs presenting genitourinary lesions not amenable to local resection via simple episiotomy and outcomes are described.

## 2. Surgical Techniques

In the next paragraph, partial vaginectomy, complete vaginectomy, partial vestibule-vaginectomy, vulvo-vestibule-vaginectomy and vulvo-vestibulectomy, as performed by the authors, will be described. All information described below is summarized in Table 1.

### 2.1. Partial Vaginectomy

Partial vaginectomy (PV) is requested when the first third of the vagina is involved in a pathological process. The main indication for a partial vaginectomy is the treatment of benign or malignant neoplasia, chronic cranial vaginitis unresponsive to medical or surgical treatment such as urethropexy, or cervix trauma occurred during parturition. In the case of malignant tumour, PV is recommended if there is no local dissemination or distant metastases. It could also be performed as a palliative procedure for advanced malignancies when the owner does not allow other surgical procedures, the tumour pushes organs such as rectum or bladder, or in elderly patients. Chronic cranial vaginitis could be consequent to congenital malformation of the vagina and vestibule that cause an urinary reflux or to an abnormal outflow of urine or urine stagnation in the vagina caused by topographic abnormalities of the genital tract following sterilization at a very young age [10].

PV is performed via an abdominal approach with the patient in dorsal recumbency. A ventral midline incision from the umbilicus to the pubis is performed. In intact or ovariectomized dogs the surgical procedure is completed in two steps: in the first step, routine ovariohysterectomy (OVH) or hysterectomy are performed and the bladder is retroflexed to expose the vagina (Figure 2).

In the second step, the cranial vagina is resected. The rectogenital and vesicogenital pouch are carefully dissected via blunt and sharp dissection with the purpose to allow the complete isolation of the vagina. The dissection must be performed very close to the vagina in order to prevent inadvertent trauma to the ureters, and to preserve the innervation of the bladder and urethra. Vascular supply (uterine artery and vein arising from the vaginal trunk) that runs laterally the cranial part of the vagina are ligated close to the vagina and close to the rectogenital and vesicogenital pouch. The vagina is then cranially pulled to allow the resection as caudally as possible. The vaginal stump is sutured with single, introflexing detached sutures. The abdominal wall is then closed as routine.

PV performed via abdominal approach presents minimal complications, but the surgeon must be careful when working between the two ureters at the base of the broad ligament lateral the uterus body and vagina: in overweight or obese dogs, the ureters could be difficult to recognize. Urethra must be recognized and not damaged as well as vaginal vascularization to avoid postsurgical complications. The use of a urinary catheter makes it easier to recognize the urethra. Moreover, in some cases, it could be necessary to resect the vagina more caudally, depending on the extension of the lesion: in this case it could be necessary to tie the vaginal artery and vein. These vessels must be ligated as close as possible to the vagina in order to avoid unintentionally tiding of bladder and urethra vascularization.

However, depending on the breed, the patient state of nutrition, neoplasia location or topographic position of the vagina, it could be necessary to break through the vesicogenital pouch that divided the peritoneal cavity from the retroperitoneal one in order to allow the complete resection of the organ involved. Moreover, the uterine and vaginal artery (arteria vaginalis, formerly urogenital artery) on the lateral wall of the vagina, the pelvic plexus (plexus pelvicus, also named hypogastric plexus) located on both side of the cervix and pudendal nerve that runs along the dorsal edge of the internal pudendal artery must be identified and preserved. It must be emphasized that in PV, due to the pubic bone, a complete visualization of the structures in this area could not be always possible, thus palpation could help their identification.

Intraoperative complications are haemorrhage, urethral damage (such as rupture or stretching) and lesion of the rectum, bladder or clitoris’ corpora cavernosa consequent to a damage of the pelvic plexus. In the post-operative period, it is sometimes possible to detect urethral spasms that could led to transitory urinary retention more frequently than urinary incontinence. Intra- and post-operative complication could be avoided or minimized if the surgery is properly performed as described and all steps are carefully executed.

### 2.2. Complete Vaginectomy

Complete vaginectomy (CV) is indicated when benign or malign tumours, chronic vaginitis or neoformations involving the entire vagina and a PV cannot be performed.

First of all, ovariohysterectomy (OVH) or hysterectomy via abdominal approach are performed in intact and ovariectomized dogs. If the dog has previously undergone ovariohysterectomy, before CV is performed, it may be necessary to solve possible adhesions between the vaginal stump, the bladder and the rectogenital and vesicogenital pouch.

CV is performed via a perineal approach with the dog in sternal recumbency. The perineal skin, from the base of the tail to the vulvar lips and laterally to the mid-thigh region, are shaved. A pursue-string suture is placed in the anus and a sterile Foley catheter is inserted in the urethra. Immediately after surgery, the pursue-string suture is removed. A midline incision between anus and the dorsal commissure of the labia of the vulva (commissura labiorum dorsalis) is made. Tissue and muscle (ischiocavernosus, ischiourethralis and superficial branch of the external sphincter ani) are carefully dissected via blunt and sharp dissection in order to allow the complete isolation of the vagina. Vascular supply to the vagina (vaginal artery and vein) is ligated as close as possible to the organ in order to prevent damage to the other vessel or nerve that could be found in this anatomical region. The isolated vagina is then retroflexed and retracted caudally. Attention must be paid to the urinary meatus, which must not be removed or damaged during the resection of the vagina: the urethra and its junction with the vaginal floor are identified via the urinary catheter and the vagina is resected just cranially in order to left enough tissue to perform an introflexing detached suture. That allowed the complete asportation of the vagina and hymen without involving the urinary meatus. The subcutaneous and cutaneous tissue are closed with attention to ureters, urethra, vessel and pelvic plexus which must not be involved in the suture. Intra- and post-operative complication are similar to those described for PV.

### 2.3. Partial Vestibule-Vaginectomy

Partial vestibule-vaginectomy (PVV) is recommended as treatment for benign or malignant neoplasia, chronic cranial vaginitis unresponsive to medical or surgical treatment, congenital and acquired malformation like stenosis, cyst or paramesonephric residues, vaginal diseases near to the hymen or in the proximal third of the vestibule.

Ovariohysterectomy (OVH) or hysterectomy via abdominal approach are previously performed in intact and ovariectomized dog.

PVV could be performed by perivaginal approach with the dog in sternal recumbency. The surgery procedure is the same as previously described for CV, but the resection of the organ includes, besides the vagina, a variable part of the vestibule depending on the extension of the lesion.

During this procedure the urethra is identified via the urinary catheter, dissected free via blunt dissection and resected at its junction with the vestibula floor. After the resection of the vagina and vestibule, dead spaces are closed by reopposing adjacent soft tissue structures. In order to perform the urethrostomy, an episiotomy is needed to expose the floor of the remaining vestibule. An incision on the midline of the vestibule floor is then performed, the spatulate cut end of the urethra is exteriorized through the small stab incision and the urethral mucosa is sutured to the vestibule mucosa with a 4-0 absorbable monofilament glycomer simple interrupted suture. As the urethrostomy is completed, the episiotomy incision will be closed with a simple continuous pattern with a braided polyglycolic acid absorbable suture in three layers: vestibula mucosa, muscles and subcutaneous tissue and lastly the skin.

Intra- and peri-operative complication are the same as described for the other surgeries. A Foley catheter must be left in situ for a couple of days to allow complete healing of the urethrostomy.

### 2.4. Vulvo-Vestibulo-Vaginectomy

Vulvo-vestibulo-vaginectomy (VVV) is indicated for benign large or multiple neoplasia involving the vagina, vulva and vestibule, malignant neoplasia, chronical vaginitis and vestibulitis.

VVV could be performed by a mixed approach (abdominal and caudal approach) or caudal approach alone: a perineal urethrostomy is always required.

The mixed approach is mandatory in intact or ovariectomized dog and in all cases with suspected abdominal adhesion. The surgical procedure is completed in two steps: an abdominal approach for ovariohysterectomy (OVH) or hysterectomy and a caudal approach for the resection of the vagina, vulva and vestibule. The surgical field is prepared as described for the perineal approach for the CV procedure.

Subsequently to OVH or hysterectomy, the bladder is retroflexed to expose the vagina and associated structures. The surgical procedure is the same as described in the PV. In the second step, the patient is placed in sternal recumbency to expose the perineal region. An elliptical incision around the vulvar margin is made, followed by the clitoral artery isolation and ligation in order to prevent emorrhagia. The caudal vagina and vestibule are freed via blunt dissection from the surrounding tissue and ischiocavernosus, ischiourethralis, the superficial branch of the external sphincter ani muscle and constrictor vestibuli muscles are resected, in order to allow a complete dissection (Figure 3). The vaginal vessel (artery and vein) are tied and cutted as close as possible to the organs: caution must be paid to not damage the urethra and its vascularization. The catheterized urethra is resected at the junction with the vaginal floor. The vagina, vestibula and vulva are then retracted caudally and removed. The deeper tissues are closed with a 2-0 absorbable braided polyglycolic acid continuous suture. In order to perform the perineal urethrostomy, the distal urethra is pulled caudally, the cut end of the urethra is spatulate to increase the luminal diameter and sutured to the skin under the anus, slightly ventral to the neck of the bladder, with a 4-0 absorbable monofilament glycomer simple interrupted suture. In a medium/large size dog, indicatively, the position of the urethrostomy is about 4 cm under the anus.

In caudal approach all the vagina, vestibule and vulva are completely isolated from the surrounding tissue and resected as described above. The vaginal artery must be ligated as deeply as possible.

In both caudal and mixed approach, the urethra must be managed with caution to prevent damage to the distal tract and must be freed from the muscular component to correctly perform the urethrostomy. If the vessel of vaginal and clitoral are not correctly identified and ligated, a haemorrhage may occur.

### 2.5. Vulvo-Vestibulectomy

Vulvo-vestibulectomy (VV) is indicated for benign clitoral neoformation, malignant not infiltrated neoplasia, vulvo-vestibular traumatic or congenital lesions such as labial aplasia or vulvar stenosis.

The patient must be prepared as described for the caudal approach of the CV. An elliptical incision around the vulva is performed, followed by the clitoral artery isolation and ligation. Vulva and vestibule are dissected via blunt and sharp dissection from the surrounding tissue and the ischiocavernosus, ischiourethralis and superficial branch of the external sphincter ani muscles are resected in order to allow vestibula resection just caudally of the urinary meatus. Subsequentially, a restriction of the remnant part of the vestibula diameter, the stump of the genital tract will be pulled caudally and inclined in order to prevent an incorrect urinary out flow or vaginal urinary stagnation: the vestibule mucosa is then sutured to the skin with simple interrupted suture.

The technique does not require particular attention, except to avoid injuring urethra and urinary meatus.

## 3. Materials and Methods

The medical records of thirty-three female dogs that underwent PV, CV, PVV, VVV or VV from 2008 to 2021 were reviewed. Age, breed, medical history, clinical findings, preoperative diagnostics, post operative management, complications occurred during the surgery and in the postoperative period, and histological diagnoses were recorded.

In all cases where a vaginal mass was suspected, diagnosis has been made on the basis of history and clinical evaluation that include vaginal and rectal palpation: in some cases, biopsy or fine-needle cytology of the mass were performed. Ultrasonography of the reproduction tract (Esaote MyLab Vet5, probe micro-covex 9-3 Mhz), biochemical and haematologic profiles, abdominal ultrasound examination (Philips Epiq 5G, different probe has been use depending on the patient size: micro-convex C8-5, covex C9-2 or linear L5-12) and thoracic radiographic (Mercury-332, three projections) for cancer staging (when necessary), vaginal cytology was performed in all bitches.

### 3.1. Anesthetic Protocol

In each bitch, after a premedication with an intramuscular injection of 0.01–0.02 mg/kg acepromazina (Prequillan^®^, Ati PETS, Bologna, Italy) or 0.005 mg/kg dexmedetomidina (Dextroquillan^®^, Ati PETS, Bologna, Italy) and 0.2 mg/kg metadone (Semfortan^®^, Dechra Veterinary products srl, Torino, Italy), general anaesthesia was induced by administration of 2–4 mg/kg propofol (Proposure^®^, Boheringer Ingelheim, Milan, Italy) intravenously. Anaesthesia was maintained with 1.5–2% isoflurane (IsoFlo^®^, Esteve Farma Lda, Milano, Italy) in oxygen.

### 3.2. Statistic

A descriptive statistic has been performed on the data, and percentage incidence of lesions that were identify was generated.

## 4. Results

### 4.1. Population

The medical records of 354 female dogs referred for vaginal problems to the University Veterinary Hospital Giuseppe Gentile of the University of Bologna from 2008 to 2021 were reviewed. Of these cases, only 33 female dogs that underwent PV, CV, PVV, VVV or VV were included in the study. All intact and ovariectomized dogs underwent ovariohysterectomy or hysterectomy, except those which needed vulvo-vestibulectomy.

Twenty-three were intact bitches, nine were spayed and one had an ovarian remnant. The mean weight was 19.35 kg sd 8.46 (median weight 20 kg; range 3–31.8 kg) and the mean age was 9.39 years sd 2.96 (median age 9 years; range 3–15 years). Breeds represented included mongrel (12), Beagle (1), Boxer (3), Labrador (2), Yorkshire Terrier (2), Cavalier King Charles (1), Cirneco dell’Etna (1), Dalmatia (1), English Bulldog (1), English Setter (1) English Cocker Spaniel (1), French Bulldog (1), German Shepherd (1), Golden Retriever (1), Irish Setter (1), Maremman Shepherd (1), Pointer (1) and Siberian Husky (1). Signalment data of the patients are reported in Table 2.

### 4.2. Clinical Findings and Preoperative Diagnostic Tests

At presentation, as reported in Table 2, clinical signs included: vulvar discharge of different nature (12/33), dysuria (4/33) and urinary retention (2/33) as a result of the compression of the urethra by the mass, increased volume of the vulva (5/33), a prolapsed mass through the vulva (3/33), vaginitis (3/33), increased volume of the perineal area (2/33), polyuria (2/33), urinary tenesmus (2/33), anorexia/dysorexia (1/33), depression (1/33), ulcerative mass prolapsed through the vulva (1/33), fecal tenesmus (1/33) and uroperitoneum (1/33). Six dogs had no reproductive symptoms. Eleven dogs presented one or more concurrent diseases such as mammary tumour (2/11), ovarian remnants (1/11), ovarian tumour (1/11), ovarian cyst (1/11), uterine neoformation (1/11), vaginal septa (1/11), hydrometra (1/11) and cystic endometrial hyperplasia (5/11). The serum chemistry profiles were within normal limits and no evidence of metastasis was shown in the chest radiographs and abdominal ultrasound at the moment of the surgery in all dogs except for dog 26. Cytology examination was performed in nine dogs, whilst 3 underwent a preoperative biopsy examination. The results are listed in Table 2.

Dog 26 arrived to our clinic for an uroperitoneum originated from a vaginal rupture following surgical closure of the urinary meatus following the surgical removal of a vaginal mass.

### 4.3. Surgery

The surgical technique was planned in function of the clinical examination, abdominal ultrasonography, vaginoscopy and/or vaginography performed and the direct intraoperative visualization of the neoplasia and/or cytological/biopsy preoperative evaluation.

Thirty dogs underwent surgery for removal of a vaginal or vulvar neoformation, two for a fluid filled vagina, and one for a fluid filled vagina associated to a chronic vaginitis. PV was performed in 14 dogs, CV in one dog, PVV with a vestibule urethrostomy in two dogs, vulvo-vestibule-vaginectomy with a perineal urethrotomy in seven dogs and VV in nine dogs. Ovariohysterectomy was performed in all intact dogs that underwent PV or CV and VVV. In two dogs who had undergone, respectively, PVV and PV and were previously neutered by ovariectomy, hysterectomy was performed.

Surgery was successfully achieved and well tolerated in all dogs.

### 4.4. Postoperative Management

An urinary catheter was left in place for 24–48 h in VV and up to 7 days in VVV or PVV; a petroleum jelly was applied to the perineum to prevent urine scalding at least for 10 days. An Elizabethan collar was used until the removal of the suture. Pain was managed with the administration of 2–4 mg/kg tramadol every 8–12 h (Altadol^®^, Formevet, Milano, Italy) and 2–4 mg/kg carprofen (Rimadyl^®^, Zoetis Italia S.r.l, Roma, Italy) or 1 mg/kg robenacoxib (Onsior^®^, Elanco Italia S.p.a, Sesto Fiorentino, Italy) once a day for 3 days to reduce inflammation and edema. Amoxycillin/clavulanic acid (Clavaseptin^®^, Vétoquinol Italia S.r.l., Bertinoro, Italy) at 12.5 mg/kg twice a day for 10 days was administered as prophylaxis.

Histopathological examination was always performed except when the owners refused to do it.

Hospitalisation after surgery was between 24–48 h and was necessary for the management of postoperative pain and the urethral catheter. Only in two cases (dog 12 and dog 26) was hospitalization longer. In dog 12, who underwent vulvo-vestibule-vaginectomy and ovariohysterectomy because of neoformations in cranial vagina and in the vulva, the surgery was quite long and a severe haemorragia was observed, requiring a longer recovery due to pain management and blood transfusion. In dog 26, the longer hospitalization was according to clinical alteration referred to the uroperitoneum presented before surgery. In all cases, except for two (dog 1 and 12), the short-term outcome of 7 days was without complications. No micturition or defecation problems were observed, nor ureters or bladder anomalies were ultrasonographically detected in post-operative stage. In dog 1, as a large skin excision lateral the urethrostomy area was necessary, the healing process led to a deviation of the urethrostomy site and a consequent deviation of the urine jet was observed.

The postoperative management of dog 26 was without complication, even if in this case surgery was a palliative cure, as metastases were observed at the presentation.

### 4.5. Histologic Diagnosis

Twenty-eight samples were analysed via histopathology; in five cases, the owners refused the histological examination. Vaginal neoplasia (25/28), an intramural cyst (2/28) and a fluid filled vagina with a severe chronic purulent ulcerative vaginitis (1/28) were diagnosed. The results of the histological or cytological examination, performed during the preoperative evaluation, has been confirmed by the histopathological examination done after surgery excepted for dog 33 where the cytological examination was in doubt because the poor cellularity. The results of the histopathological examination are shown in Table 3 and revealed leiomyoma (8/2), fibroma (6/28), carcinoma (2/28), myofibroblastic fibrosarcoma (2/28), intramural vaginal cyst (2/28), mast cell tumour (2/28), fibrosarcoma (1/28), fibromyosarcoma (1/28), undifferentiated carcinoma (1/28), extra intestinal gastrointestinal stromal tumour (1/28), canine transmissible venereal tumour (1/28), leiomyosarcoma (1/28) and severe chronic purulent ulcerative vaginitis (1/28). Twenty-five tumours were processed: 13 resulted as malignant and 12 benign.

## 5. Discussion

Aim of this article was to describe five different surgical techniques to correct diseases affecting the vaginal, vestibular and vulvar tract in the bitch. To date, the literature reports only four surgical techniques involving this reproductive tract: the authors instead differentiate five techniques depending on the portion that has to be removed and gave instructions to plan the best and less demolishing intervention in agreement with the disease, the age and the physical condition of the patient. Of the five surgical techniques we have described, in the literature, no cases of vulvo-vestibulectomy (removal of the vulva and vestibule tract where the urinary meatus can or cannot be touched) are reported. The surgical techniques depicted in this essay are classified in function of the anatomical female reproductive tract involved in the surgery as illustrated in Figure 1.

The main problem we had to face, related to surgical techniques described in the literature, is linked to the nomenclature, to not very detailed description of different approaches in the literature, and to the indications relating to the sections of the apparatus that have to be removed. Due to these unclear aspects, it was not easy to understand what type of intervention was performed in the literature and correctly compare the techniques and the results obtained.

Surgical textbooks [11], reviews [2,4] as well as case reports [3,7,12,13,14,15] used similar or same names to refer to surgery involving vagina, vestibule and vulva, but they did not detail what exactly had been removed. This implies that to date the literature provides the reader with unclear and approximate information. Furthermore, this greatly complicates the possibility of making comparisons between the techniques described by the authors and our results.

As regards demographic aspect, our data are in line with what has been found in the literature: adult bitches (average age 9 years), mainly whole dogs with neoplastic concurrent diseases [3,7,12,13,15]. Additionally, the symptoms are comparable to what is reported by other authors. In younger bitches, average age 3,6 years (case 5, 14 and 28), the signs observed are compatible with alterations of the vaginal tract found in young bitches. In particular, case 14 is similar to the case described by Folk et al. [16], who performed a partial vaginectomy for chronic cervico-vaginitis. In accordance with the colleagues, it is the authors’ opinion that this surgical technique can be considered for chronic vaginitis that do not respond to medical treatments, as it is curative and well supported by the patient.

In our cases, 13 of the 25 tumours analysed histologically were malignant, and 12 were benign. This result seems to be in contrast with data concerning the incidence of vaginal tumours in the literature as it was reported that benign tumours show a higher incidence than malignant tumours [5,6,17,18]: on the other hand, if we consider all vaginal neoplasms referred in the same period, we found that 85.5% were benign neoplasms and 14.5% were malignant. We hypothesize that our data were different from the literature, as we deal with cases referred by colleagues for a specific surgical approach.

The 5 techniques described were well tolerated by the patients and a complete recovery and hospitalization was observed within 24–48 h in accordance with what is described in the literature [3,14,15]. Post-operative management was normal (pain managed with anti-inflammatory in older cases, with methadone and buprenorphine in the most recent) [3,7,14,15]. The hospitalization was linked to the management of the urinary catheter or for pain management. Only two cases had a hospitalization longer than 3 days related to pre-surgical clinical conditions: case 26 (French bulldog) presented uroperitoneum at the entrance and case 12 (golden retriever) needed a transfusion in the postoperative period due to massive intraoperative blood loss. An excessive bleeding was observed in 2 of the 11 cases presented by Nelissen et al. [3], but did not require post-operative transfusion. In none of the 33 cases described, regardless of the technique used, were important intraoperative complications found, except for the massive intraoperative blood loss yet described in case 12 and due to the important vascularization linked to the clitoral mass.

In the post-operative period (short term outcome 7 days), we did not find any complications affecting the urinary tract and colon/rectum. Two cases of transient urinary incontinence (case 1 and 6) which resolved spontaneously in a couple of days were observed (as described also in literature [3]) and one urinary retention lasting about 5 days, which resolved spontaneously, apparently linked to the inflammation developed at the surgery site (case 8) (as described also in literature [12]). Furthermore, in the postoperative period, an extensive skin removal was observed at the urethrostomy site in dog 1: the skin healing process resulted in a malposition of the urethrostomy site with consequent deviation of the urine flow.

The minor post-operative complications observed in our cases are in agreement with what is described in the literature [3,12,13,14,16,19].

Ogden et al. [7] and Nelissen [3] reported a non-transient urinary incontinence as one of the major complications of these surgical techniques, nevertheless, in the 33 cases described in this article, it has never been encountered. These authors justified this complication with the more invasive surgery, causing damage to the pelvic plexus and the innervation of the region, and/or the presence of a pelvic bladder, known to be one of the causes of post castration urinary incontinence [3,7]. These authors argue that the loosening of pelvic attachments due to dissection in the region of the bladder could also theoretically cause a more intrapelvic bladder position and, consequently, contribute to subsequent urinary incontinence [3]. We think that both hypothesis are plausible, but the latter is more plausible than the previous one, and it is true that it would be useful to verify by abdominal radiography the presence of a pelvic bladder in subjects who have urinary incontinence following surgery in order to better validate this hypothesis. Furthermore, in the authors’ experience, no case of permanent urinary incontinence has been detected, and it is our opinion that an accurated and more “targeted” surgery could reduce this risk.

Some authors consider these very specialized and very invasive/radical surgical techniques and therefore advise against them [2,4], while others advise against it only for malignant tumours due to the risk of recurrence or tumours that have already metastasized [15]. Some authors [12,13,14] consider these techniques useful in order to remove large neoformations or benign tumours, as present minor and often transient post-operative complications (mild incontinence, infection urinary tract, difficulty in urination). Therefore, these techniques could be curative in case of benign or malignant tumours that have not yet metastasized, and palliative in other diseases. Moreover, we argue that these surgeries are also useful for therapeutic purposes for other diseases, such as chronic vaginitis, that do not respond to traditional treatments, or for vaginal cysts, as claimed by Folk et al. [16], but they can also be used for treatment of congenital vaginal abnormalities.

Although CT scans were not performed, as the owners did not consent, it is the authors’ opinion that performing advanced diagnostic tests may be useful in order to better establish the margins of the lesion and the degree of the infiltration: however, in all 33 cases described in the article (with the exception of case 26, which we already knew had metastases), surgery was 100% resolutive and the margins decided on the basis of the preoperative and intraoperative diagnostics were sufficient. In reality, among 44 cases described in the articles mentioned, only 12 cases underwent preoperative CT (1 case of [19] and 11 of the 21 cases described by [7]). Furthermore, Igna et al. [14] in 2016 did not use CT or MRI in the preoperative and underlined that “postoperative examination of the resected tumor confirmed that it had been totally excised”, which is comparable to what we claim.

It is our opinion that having five different available techniques to approach vaginal neoplasms depending on their size, location and nature is useful in order to perform the best surgery according to the patient characteristics, technique invasiveness and whether it is palliative or not, as sometimes we had to improve only the quality of life, so it may be appropriate to choose a technique rather than another less invasive and destructive one.

Moreover, in our opinion and experience, the possibility to perform an urethrostomy at the vestibular level instead of percutaneous urethrostomy presented several advantages in order to protect the urinary tract from infections or trauma. Finally, nowadays even in veterinary field, the importance of aesthetic factors must be considered, for example, when we leave the vulva.

The main limitation of our study is related to the kind of surgical techniques described, which are rarely needed and performed in practice. Therefore, due to the number of patients, it was not possible to perform an adequate statistical analysis or identify particular predispositions within the study population.

Another limitation is represented by the difficulty to compare our data with those previously published, as the literature reports little information and an ambiguous description of these surgical techniques.

## 6. Conclusions

We described in detail five different surgical techniques available for the female copulatory organs related to a correct anatomical nomenclature. All techniques here described results well tolerated by the patient. In the authors’ opinion, the possibility to choose between five different techniques according to the portion of the female reproductive tract that has to be removed and the possibility to plan the best and less demolishing intervention in agreement with the disease, the age and the physical condition of the patient are important points.

## Figures and Tables

**Figure 1 animals-12-00196-f001:**
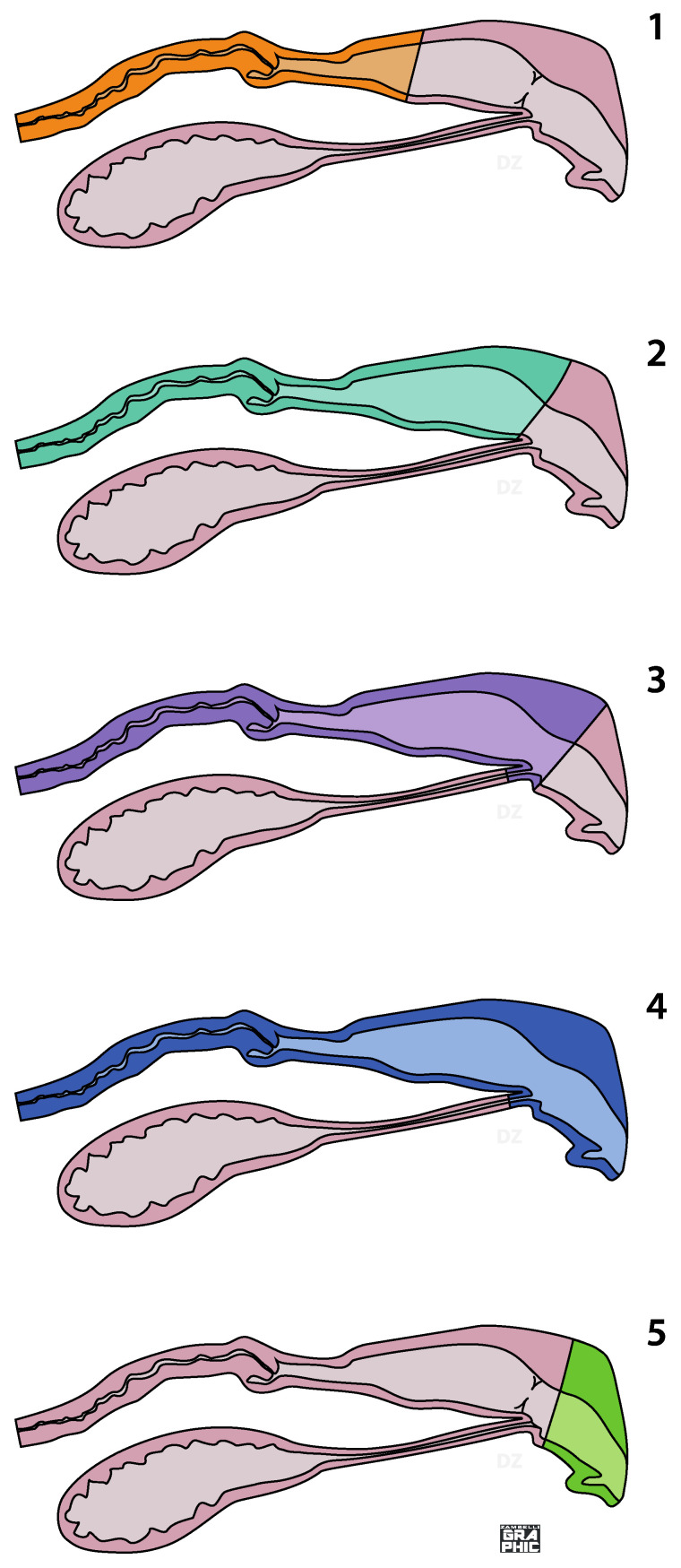
Schematic presentation of the five types of surgery: (1) partial vaginectomy (PV), consists in the ablation of the cervix and a variable length of the first part of the vagina; (2) complete vaginectomy (CV), all vaginal tissue, from the cervix to the hymen, are removed; (3) partial vestibule-vaginectomy (PVV), is similar to the complete vaginectomy, but a variable part of the vestibule is involved; (4) vulvo-vestibule-vaginectomy (VVV), consists in the ablation of the last part of the reproduction tract of the bitch, from the vulva to the cervix; (5) vulvo-vestibulectomy (VV), the externa genitalia and a variable vestibular length is removed.

**Figure 2 animals-12-00196-f002:**
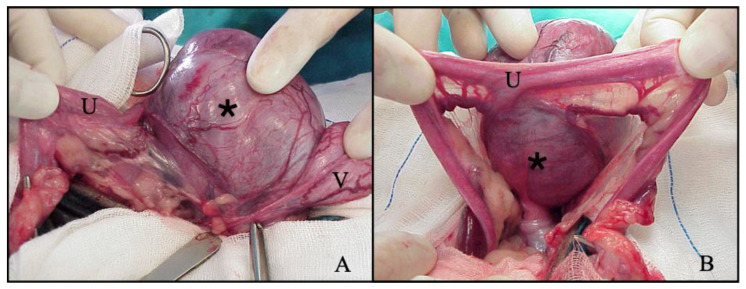
Intraoperative image of neoformation in cranial vagina removed by partial vaginectomy. (**A**): lateral view; (**B**): sagittal view. (U): uterus; (*): vaginal neoplasia; (V): bladder.

**Figure 3 animals-12-00196-f003:**
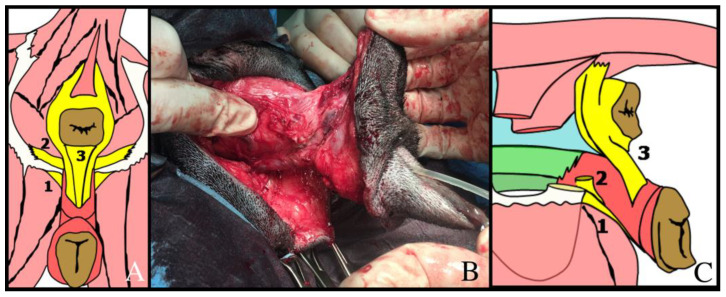
Caudal (**A**) and lateral (**C**) schematic presentation of the perineal muscles: (1) ischiocavernosus muscle; (2) ischiourethralis muscle; (3) vestibular muscle. (**B**) Intraoperative image of the lateral view of vulva and vestibule after the resection of the ischiocavernosus and ischiourethralis muscles.

**Table 1 animals-12-00196-t001:** Principle information about the five surgical techniques described in the study.

Surgical Technique	Description	Indication	Surgical Approach	Ovariohysterectomy Hysterectomy	Urethrostomy	Intraoperative Complications
Partial Vaginectomy (PV)	Removal of a variable tract of the vagina, always including the cervix	Involvement of the first third of the vagina by pathological process	Abdominal	Yes	No	Hemorrhage, urethral damage, lesion of the rectum, bladder or clitoris’ corpora cavernosa consequent to a damage of the pelvic plexus
Complete Vaginectomy (CV)	Removal of all vaginal tissue, from the cervix to the hymen	Tumours, chronic vaginitis, and cases where PV cannot be performed	Perineal	Yes	No	Similar to those described for PV
Partial Vestibule-vaginectomy (PVV)	Similar to the CV but a variable part of the vestibule is involved	Neoplasia, chronic cranial vaginitis unresponsive to medical or surgical treatment, congenital and acquired malformation, vaginal diseases near to the hymen or in the proximal third of the vestibule	Perivaginal	Yes	Yes	Similar to those described for PV
Vulvo-vestibulo-vaginectomy (VVV)	All the copulatory organs (vagina, vestibule and vulva) are removed	Large or multiple neoplasia involving vagina, vulva and vestibule, chronical vaginitis and vestibulitis	Mixed or Perineal	Yes	Yes	Similar to those described for PV
Vulvo-vestobulectomy (VV)	The externa genitalia and a variable vestibular length is removed, the urethral tubercle is not involved	Benign clitoral neoformation, malignant not infiltrated neoplasia, vulvo-vestibular traumatic or congenital lesions	Perineal	No	No	The technique does not require particular attention except to avoid injuring urethra and urinary meatus

**Table 2 animals-12-00196-t002:** Signalment and symptoms presented at the first visit, cytological evaluation of the lesion and concurrent disease.

Case	Breed	Age [Years]	Weight [Kg]	Reproductive State	Symptom	Preoperative Cytology of the Lesion	Preoperative Histopathology of the Lesion	Concurrent Disease
1	Mongrel	10	12	neutered	no symptom	no	no	
2	Dalmata	8	18	intact	vulvar discharge (haematic)	no	no	CEH, vaginal septa
3	Mongrel	11	16	ovarian remnant	no symptom	no	no	ovarian remnant
4	Yorkshire Terrier	13	3	intact	vulvar discharge (haematic)	no	no	CEH
5	Mongrel	4	26.9	neutered	vulvar discharge (haematic)	Not diagnostic	no	
6	Engish Cocker Spaniel	9	14	intact	vulvar discharge (purulent), increase volume of the vulva	no	Fibroma	
7	Maremma Shephers	11	28	intact	vulvar discharge (purulent), increase volume of the perineal area	no	Fibroma	udder tumour
8	Boxer	8	25	neutered	increase volume of the vulva	no	Myofibroblastic fibrosarcoma	
9	Mongrel	13	14	neutered	vulvar discharge (haematic), urinary retention	no	no	vulvar neoformation
10	Cirneco dell’Etna	6	10	intact	increase volume of the perineal area	no	no	
11	Mongrel	10	24	neutered	prolapsed mass through the vulva	Epithelial neoplasia, compatible with a mesenchymal neoplasia with malignant characters	no	
12	Golden Retriever	14	31	intact	polyuria, faecal tenesmus	Clitoris FNA not diagnostic, abdominal mass FNA compatible with a mesenchymal neoplasia with malignant characters (suspect sarcoma)	no	uterine neoformation
13	Mongrel	7	17.3	intact	increase volume of the vulva	Epithelial neoplasia with glandular origin (suspect clitoris adenocarcinoma)	no	
14	English Bulldog	4	25	intact	vulvar discharge	no	no	Ovarian tumour (dx), ovarian cyst (sx), CEH
15	Siberian Husky	12	21.5	intact	no symptom	spindle cells	no	
16	Irish Setter	12	27	intact	no symptom	no	no	
17	Pointer	8	17	intact	urinary retention for compression of the urethral	no	no	CEH
18	Mongrel	9	14	intact	vulvar discharge, purulent vaginitis	no	no	CEH
19	English Setter	7	22	intact	increase volume of the vulva	no	no	
20	Boxer	6	25	intact	vulvar discharge, dysuria, vaginitis	no	no	
21	German Shepherd	8	27	neutered	prolapsed mass through the vulva	no	no	
22	Mongrel	6	22	intact	dysuria for urethral compression	no	no	
23	Mongrel	12	20	intact	dysuria for urethral compression	no	no	
24	Beagle	12	10	neutered	vulvar discharge, prolapsed mass through the vulva, ulcerate neoformation	no	no	
25	Mongrel	9	16	neutered	vulvar discharge, vaginitis	no	no	
26	French Bulldog	9	11.9	neutered	uroperitoneum	no	no	
27	Labrador	9	31.8	intact	no symptom	mast cell tumour	no	
28	Cavalier King Charles	3	7.6	intact	vulvar discharge (mucosal)	compatible with cist	no	
29	Yorkshire Terrier	12	3.5	intact	depression	spindle cells	no	
30	Mongrel	12	7	intact	anorexia/dysorexia, polyuria, urinary tenesmus, compression of the urethral	no	no	Hydrometra
31	Boxer	10	30	intact	dysuria, urinary tenesmus	no	no	
32	Labrador	15	26	intact	increase volume of the vulva, prolapsed mass through the vulva	no	no	
33	Mongrel	11	35	intact	no symptom	sarcoma	no	Addison, udder tumour

**Table 3 animals-12-00196-t003:** Type of surgery and histopathological findings.

Case	Type of Surgery	Urethrostomy	Concomitant Surgery	Histopathologic Diagnosis	Day of Hospitalisation
1	Vulvo-vestibule-vaginectomy	perineal	/	leiomyosarcoma	1
2	Partial Vaginectomy		Ovariohysterectomy	fibromyosarcoma	15
3	Partial Vaginectomy		Hysterectomy	/	1
4	Vulvo-vestibulectomy		/	/	0
5	Vulvo-vestibulectomy		Hysterectomy	TVT	1
6	Vulvo-vestibule-vaginectomy	perineal	Ovariohysterectomy	fibroma	1
7	Vulvo-vestibule-vaginectomy	perineal	Ovariohysterectomy	fibroma	1
8	Vulvo-vestibule-vaginectomy	perineal	/	myofibroblastic fibrosarcoma	0
9	Partial Vaginectomy		/	leiomyoma	2
10	Complete vaginectomy		Ovariohysterectomy	vaginal cyst	0
11	Vulvo-vestibulectomy		/	extra intestinal gastrointestinal stromal tumour	0
12	Vulvo-vestibule-vaginectomy	perineal	Ovariohysterectomy	undifferentiated carcinoma and vaginal polyp	3
13	Vulvo-vestibulectomy		/	/	0
14	Partial Vaginectomy		Ovariohysterectomy	severe chronic purulent ulcerative vaginitis	0
15	Partial Vaginectomy		Ovariohysterectomy	leiomyoma	0
16	Partial vestibule-vaginectomy	vestibule	Ovariohysterectomy	fibroma	0
17	Partial Vaginectomy		Ovariohysterectomy	leiomyoma	0
18	Partial vestibule-vaginectomy	vestibule	Ovariohysterectomy	myofibroblastic fibrosarcoma	0
19	Vulvo-vestibulectomy			fibrosarcoma	0
20	Vulvo-vestibule-vaginectomy	perineal	Ovariohysterectomy	/	0
21	Vulvo-vestibulectomy		/	carcinoma	0
22	Partial Vaginectomy		Ovariohysterectomy	leiomyoma	0
23	Partial Vaginectomy		Ovariohysterectomy	fibroma	0
24	Vulvo-vestibulectomy		/	carcinoma	0
25	Partial Vaginectomy		/	leiomyoma	0
26	Vulvo-vestibulectomy		/	mast cell tumour	3
27	Vulvo-vestibulectomy		/	mast cell tumour	0
28	Partial Vaginectomy		Ovariohysterectomy	vaginal cyst	0
29	Partial Vaginectomy		Ovariohysterectomy	leiomyoma	0
30	Partial Vaginectomy		Ovariohysterectomy	leiomyoma	2
31	Partial Vaginectomy		Ovariohysterectomy	fibroma	2
32	Vulvo-vestibule-vaginectomy	perineal	Ovariohysterectomy	/	5
33	Partial Vaginectomy		Ovariohysterectomy	leiomyoma and fibroma	3

## Data Availability

Data generated or analysed during this study are included in this published article. The raw datasets used and analysed during the current study are available from the corresponding author on reasonable request.

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
