# Peer review of "Partial Vaginectomy, Complete Vaginectomy, Partial Vestibule-Vaginectomy, Vulvo-Vestibule-Vaginectomy and Vulvo-Vestibulectomy: Different Surgical Procedure in Order to Better Approach Vaginal Diseases"

_animals, 2022, doi:10.3390/ani12020196_

Round 1
Reviewer 1 Report
The work describes the surgical approach for the resolution of pathologies of the vulvovaginal tract in the bitch. In the introduction, the possible surgical options of the vulvovaginal tract are reported in great detail and the authors propose a more precise classification of the applicable surgery techniques based on the anatomical tract(s) that need(s) to be removed. In the materials and methods and results sections, the authors report data obtained from their clinical case history, describing for each case clinical symptoms, preoperative diagnosis, post-surgical management and possible complications.
The manuscript is a hybrid between a review study and a case contribution. The author’s intention is interesting because these kind of interventions are rarely performed and the literature on this subject is rather limited. Therefore, I suggest reorganizing the manuscript as a surgical case contribution integrating the available literature with the classification proposed by the authors. The discussion is redundant and should be limited to the most important findings with relevant comparisons with the relevant literature. I suggest to better describe the outcomes of the different surgery techniques. This would add value to the paper and would represent a useful guide to address the surgeon’s choice.
Introduction
Lines 57-69. This paragraph provides a brief description of the organs of the female genital apparatus with reference to the NAV. I think this description if not necessary. The citation of the NAV is enough. The descriptions are widely reported in any anatomical book and does not need to be added to a scientific paper.
Lines 42-43 Change to ‘For these pathologies, surgical management could require simple or extremely complex approaches’
Figure 2. Neoformation of the cranial vagina removed by cranial vaginectomy: intraoperative image. A: lateral view of the uterus (U), vaginal neoplasia (*) and bladder (B); B: sagittal view of the 120 same case of the uterus (U) and vaginal neoplasia (*)
Change to: Figure 2. Intraoperative image of neoformation in cranial vagina removed by partial vaginectomy. A: lateral view; B: ventral view.
(U): uterus; (*): vaginal neoplasia; (B): bladder.
- Surgical techniques
It is not clear if the description of the surgical techniques is intended as a review report or as original information. No citation is reported in this section and so I imagine it just describes authors’ own procedures. In this case, this part cannot be considered as a review work. If this is intended as original research work, it must be specified what is really new and what is already part of the surgical practice. The way this section is reported is typical of a book or monography destined to students or to practitioners rather to a scientific publication.
The association with other surgeries, such as ovariohysterectomy or hysterectomy, does not need to repeatedly mentioned.
- Material and methods
Some parts, such as anaesthetic protocols and routine postoperative medical management, are redundant and may be removed or synthesized.
- Discussion.
This section is detailed to the point that is hard to read. I suggest to synthesise lines 390-427.
Lines 472-477: the reference to prostatectomy sound unappropriated and I would remove it. Logically, the more invasive is the surgical technique, the more complications should be expected.
The paragraph 518-521 is not relevant and should be removed or better related to the reported results.
4.1 Population
Lines 290-296: Data of dogs are described in Table 1 so it is not necessary to repeat them in the text.
In Table 1, the term ‘sex’ in the fifth column should be changed in ‘reproductive state’
4.2. Clinical findings and preoperative diagnostic test
Lines 301-314: The clinical findings and histopathological investigations are already described in table 1 so that the detailed description is redundant.
4.3 Surgery
Lines 327-330: The surgical criteria should be described at the beginning of the paragraph
The surgeries techniques should be defined with the respective acronyms throughout the text.
Bibliography
The bibliography must comply with the journal criteria; furthermore, some citations are incomplete and must be integrated as requested by the editorial staff (nos. 1; 8; 9)
Author Response
Dear Reviewer,
Thanks for your observation and recommendations. We done quite all the revision you indicated. Unfortunately, it was not possible to make all the corrections you suggested because some of them were in contrast with the ones proposed by the other reviewer, therefore we defer to the editor decision about them.
Please see the attachment for the post-by-point response.

Reviewer 2 Report
General comments:
This manuscript approaches a very interesting thematic and with relevance for the clinical practice. It is well written and very detailed as a revision manuscript, however the main point for me is your study, as an observational study that allows a descriptive statistics for the elaboration of results with the need for a more pragmatic and clear discussion. The discussion, although with an excellent writing, should be more clear and directed to the main points.
The Material and methods section is the weakest part of the manuscript, followed by the statistics and the results presentation that should be improved.
As for the introduction is very well documented, however a suggested that it should be shortened, for example with the use of tables, as mentioned in the specific comments.
Also the main relevance for an observational study is to determine if it is a safe, tolerable, feasible and repeatable method. I think this characteristics are all presented in your study, so you should mention them in the objectives of the study and in the conclusion section.
Specific comments:
- Line 86: You should refer "33 female dogs".
- Line 97: All techniques are very well explained, however I think it would be better for the readers to have a summarized table describing the different techniques, in regard to indications, surgical approach, complications, particularities, post-surgical management.
- Line 101: Abbreviations should be used in the text but not in the main titles. The same for the other titles that follows.
- Line 135: You probably should mention here the need for urinary catheter in order to make it easy to recognize.
- Line 155: After complications you should mention some measures to implement but in a very summarized way.
- Line 209: Again the abbreviation should not be in title, so the paragraph should initiate with the name of the procedure followed by the abbreviation.
- Line 269: In regard to the biopsy of the mass, it should be described the procedure. Also, as for the ultrasonography, brand of the device, the deep and other parameters should be mentioned. The same specificities should be described for the thoracic radiophonic (device, how many projections, etc).
- Line 270: You should specify what is the standard biochemical and hematologic profiles?
- Line 273: As for the anesthetic protocol, there was no anesthetic recruitment performed during surgery? And what about antibiotics 20 minutes before surgery and 2 hours after?
- Line 291: You should mention the SD and not the +/- symbol (Lang and Altman, 2013).
- Line 297: The type of distribution of the sample population should be mentioned, if it is normal or not, regarding age and weight, you could also include and histogram if that is the case.
- Line 298: Table one and table 2 are huge. They should be considered attachments and not displayed in the main manuscript, you should try to summarized and simplified them in a more aesthetic way. For example, all qualitative binomial categories (the concurrent disease and others), should be categorized as yes or no, and you could include percentages of intact and neutered dogs in a paragraph on the text, as well as talk about the cytology of the lesion.
- Line 335: For how many days this medications was prescribed? And why not consider other medications such as CBD, it would be interesting to talk about other alternatives in the discussion.
- Line 343:Again, as for the postoperative pain management performed, do you use any type of continuous rate infusions (fentanyl, methadone, etc), for the first hours? Please specify in case you do.
- Line 372: The same advice for this table, as given for table 1.
- Line 521: The limitations of this study are missing at the end of the discussion section.
Author Response
Dear Reviewer,
Thanks for your observations and recommendations. We done quite all the revisions you indicated. Unfortunately, it was not possible to make all the corrections you suggested because some of them where in contrast with the ones proposed by the other reviewer, therefore we defer to the editor decision about them.
Please see the attachment for the post-by-point response.

Round 2
Reviewer 2 Report
We feel all major suggestions were performed.